# SVRG and Beyond via Posterior Correction

## Abstract

Stochastic Variance Reduced Gradient (SVRG) and its variants aim to speed-up training by using gradient corrections, but have seen limited success in deep learning. Here, we show surprising new foundational connections of SVRG to a recently proposed Bayesian method called posterior correction. Specifically, we show that SVRG is recovered as a special case of posterior correction over the isotropic-Gaussian family, while novel extensions are automatically obtained by using more flexible exponential families. We derive two new SVRG variants by using Gaussian families: First, a Newton-like variant that employs novel Hessian corrections, and second, an Adam-like extension that improves (continual) pre-training and finetuning of Transformer language models. This is the first work to connect SVRG to Bayes and use it to boost variational training for deep networks.

## 1 Introduction

Variance Reduction is a powerful technique to speed-up stochastic optimization. For example, stochastic variance reduced gradient (SVRG) uses full-batch gradients to stabilize future mini-batch updates (Johnson & Zhang, 2013). The method originates in the works of Roux et al. (2012); Schmidt et al. (2017); Shalev-Shwartz & Zhang (2013) which require storing individual gradients over the full dataset. Since then, a large number of variants have been proposed exploring various aspects of this method (Nguyen et al., 2017; Fang et al., 2018; Cutkosky & Orabona, 2019). Variance reduction has become a useful tool to accelerate both convex and nonconvex optimization.

Our goal here is to explore new connections of SVRG to Bayes. Currently, no work exists in this space and no foundational connections are known. We are particularly interested in investigating whether variance reduction can be effective in non-traditional settings, for example, in variational continual pre-training and fine-tuning of language models (Shen et al., 2024). Variance reduction has not yet seen a lot of success in deep learning (Defazio & Bottou, 2019). Our exploration here is meant to assess its potential in non-traditional variational training.

In this paper, we present surprising new foundational connections of SVRG to a recently proposed Bayesian method called posterior correction (Khan, 2025). Specifically, we show that SVRG is recovered as a special case of posterior correction over the isotropic-Gaussian family, while novel extensions are automatically obtained by using more flexible exponential families (Fig. 1(left)). This result is surprising because posterior correction is not a variance reduction method, rather a knowledge transfer method. The result offers a new perspective of variance reduction as knowledge transfer, for instance, through frequent mega-batch gradient computations.

Using this result, we derive new SVRG variants that go beyond existing proposals. For example, by using full-covariance Gaussians, we obtain a new variance-reduction method that employs Hessian corrections within a variational Online Newton algorithm. This differs from most works on Newton steps that only use corrections for the gradient and never for the Hessian (Derezinski, 2023; Sadiev et al., 2024; Garg et al., 2024; Sun et al., 2025). Another Adam-like extension is obtained by using diagonal covariances, implementing posterior correction over the IVON optimizer (Shen et al., 2024). Empirically, the new variant boosts IVON's speed in non-traditional settings and shows promising results at scales up to pretraining an LLM from scratch on 50B tokens; see the middle and right panels Fig. 1. In this setting, our method outperforms Adam too, which is unlike many other traditional applications where no effective gains in performance are observed. This result also far exceeds the scale of any prior work on SVRG-based methods that use mega-batches. We validate these findings with various small to large experiments on several architectures, including ResNet,

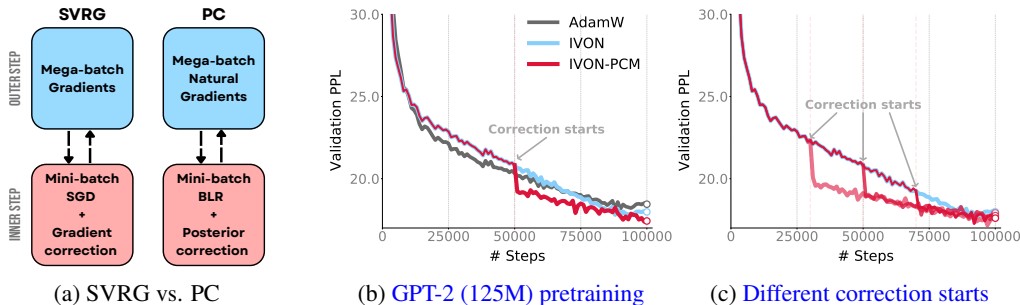

(a) SVRG vs. PC    (b) GPT-2 (125M) pretraining    (c) Different correction starts

Figure 1: (a) We present a generalization of SVRG by using Posterior Correction (PC) where gradients used in SGD are replaced by natural gradients of VB objectives via the Bayesian Learning Rule (BLR). (b) Our new IVON-PCM (red) improves performance over IVON and AdamW when pretraining GPT2-125M from scratch on ca. 50B tokens from OpenWebText. Until 50K steps IVON-PC takes the same steps as IVON, and a huge boost is obtained when correction is started. Validation Perplexities at the end are 17.4, 18.0, 18.4. (c) We show three different IVON-PC runs where correction is started at a different iteration (pink to red). We see consistent improvements irrespective of the starting iteration.

| **Algorithm 1** SVRG | **Algorithm 2** Posterior Correction (PC) |
|---|---|
| 1: Initialize $\boldsymbol{\theta}_{\text{in}}$ | 1: Initialize $\boldsymbol{\lambda}_{\text{in}}$ |
| 2: **while** not converged **do** | 2: **while** not converged **do** |
| 3:   $\mathbf{g}_{\text{out}} \leftarrow \sum_{i=1}^{N} \nabla \ell_i(\boldsymbol{\theta}_{\text{in}})$ | 3:   $\widetilde{\mathbf{g}}_{\text{out}} \leftarrow \sum_{i=1}^{N} \widetilde{\nabla} \mathcal{L}_i(\boldsymbol{\lambda}_{\text{in}})$ |
| 4:   $\boldsymbol{\theta}_{\text{out}} \leftarrow \boldsymbol{\theta}_{\text{in}}$ | 4:   $\boldsymbol{\lambda}_{\text{out}} \leftarrow \boldsymbol{\lambda}_{\text{in}}$ |
| 5:   **for** $t = 1, 2, \ldots, m$ **do** | 5:   **for** $t = 1, 2, \ldots, m$ **do** |
| 6:     Randomly pick $i \in \{1, 2, \ldots, N\}$ | 6:     Randomly pick $i \in \{1, 2, \ldots, N\}$ |
| 7:     $\mathbf{g}_{\text{in}} \leftarrow \nabla \ell_i(\boldsymbol{\theta}_{\text{in}}) - \nabla \ell_i(\boldsymbol{\theta}_{\text{out}}) + \frac{1}{N}\mathbf{g}_{\text{out}}$ | 7:     $\widetilde{\mathbf{g}}_{\text{in}} \leftarrow \widetilde{\nabla}\mathcal{L}_i(\boldsymbol{\lambda}_{\text{in}}) - \widetilde{\nabla}\mathcal{L}_i(\boldsymbol{\lambda}_{\text{out}}) + \frac{1}{N}\widetilde{\mathbf{g}}_{\text{out}}$ |
| 8:     $\boldsymbol{\theta}_{\text{in}} \leftarrow \boldsymbol{\theta}_{\text{in}} - \eta \mathbf{g}_{\text{in}}$ | 8:     $\boldsymbol{\lambda}_{\text{in}} \leftarrow (1-\eta)\boldsymbol{\lambda}_{\text{in}} - \eta N \widetilde{\mathbf{g}}_{\text{in}}$ |
| 9:   **end for** | 9:   **end for** |
| 10: **end while** | 10: **end while** |

Figure 2: Pseudo-code for SVRG (left) and our Bayesian generalization using PC (right). The latter replaces all instances of $\boldsymbol{\theta}$ and $\nabla \ell_i$ in SVRG by the natural parameter $\boldsymbol{\lambda}$ (of the posterior $q$) and natural gradient $\widetilde{\nabla}\mathcal{L}_i$. It also uses the BLR for the inner loop in line 8. We show that these differences disappear with an isotropic Gaussian $q$ and SVRG is recovered as a special case of PC.

GPT and ViT. Overall, our work encourages further investigation of the new connection to obtain real improvements with SVRG in deep learning.

## 2 BACKGROUND ON SVRG AND VARIATIONAL BAYES

In SVRG, the goal is to minimize an empirical risk $\sum_{i=1}^{N}[\ell_i(\boldsymbol{\theta})/N]$ averaged over losses $\ell_i$ for examples $i = 1, 2, \ldots, N$. A regularizer, denoted by $\ell_0$, is often added and handled by redefining the losses as $\ell_i + \ell_0/N$. Standard stochastic gradient descent (SGD) steps use stochastic gradients, for instance, the following update where the $i$'th example is randomly sampled at each iteration,

$$\boldsymbol{\theta} \leftarrow \boldsymbol{\theta} - \eta \nabla \ell_i(\boldsymbol{\theta}). \tag{1}$$

SVRG reduces the variance of such steps by using an outer loop where full-batch gradients are computed at parameters $\boldsymbol{\theta}_{\text{out}}$, and then parameters $\boldsymbol{\theta}_{\text{in}}$ are updated using stochastic increments at a randomly sampled $i$ at every iteration,

$$\boldsymbol{\theta}_{\text{in}} \leftarrow \boldsymbol{\theta}_{\text{in}} - \eta \left[ \nabla \ell_i(\boldsymbol{\theta}_{\text{in}}) - \nabla \ell_i(\boldsymbol{\theta}_{\text{out}}) + \frac{1}{N} \sum_{j=1}^{N} \nabla \ell_j(\boldsymbol{\theta}_{\text{out}}) \right]. \tag{2}$$

These steps can be seen as a full-batch gradient descent where old gradients $\nabla \ell_i(\boldsymbol{\theta}_{\text{out}})$ are 'corrected' by adding the fresh new gradients $\nabla \ell_i(\boldsymbol{\theta}_{\text{in}})$. The use of the full-batch gradient can reduce variance and speed up future mini-batch steps. Pseudo-code is shown in Alg. 1 where $m$ steps are taken in the inner loop and a constant learning rate $\eta$ is used.

SVRG is built upon earlier ideas in Stochastic Average Gradient (SAG) (Roux et al., 2012; Schmidt et al., 2017) and Stochastic Dual Coordinate Descent (SDCA) (Shalev-Shwartz & Zhang, 2013) methods where the full batch gradient is stored for all examples separately and entries are refreshed whenever corresponding examples are chosen during updating. Subsequently, many new practical variants have been proposed, for instance, SARAH (Nguyen et al., 2017) and SPIDER (Fang et al., 2018), among many other proposals (Dubois-Taine et al., 2022; Lei et al., 2017; Babanezhad Harikandeh et al., 2015). Instead of full-batch, it is also possible to use large *mega*-batches which can be, for example, 10-50 times bigger than the mini-batches. It is also helpful to use Adam and down-weight the corrections (Yin et al., 2025). There is a large number of papers that show variance reduction is useful for accelerating both convex and nonconvex optimization.

Despite this, variance reduction has not seen success in deep learning yet. The ineffectiveness is studied extensively by Defazio & Bottou (2019) who find little to no gain in run-time for traditional deep learning. They show that variance can even increase if mega-batch gradients are not refreshed often, in the end giving no effective gain in speed. The matter is complicated by the presence of additional deep learning tricks, such as, mini-batching, momentum, learning-rate schedules, etc. Numerous works have suggested new improvements but SVRG's ineffectiveness is not fixed yet (Yin et al., 2025; Tondji et al., 2021; Cutkosky & Orabona, 2019; Arnold et al., 2019; Ma & Yarats, 2019).

Clearly, using SVRG to get a real speed-up in deep learning is not easy. Here, we ask if the same is true for non-traditional settings. We consider variational training of deep networks where the IVON optimizer has lead to promising results. Such Bayesian approaches hold promise to facilitate continual, federated, and active learning of deep networks. Use of mega-batches can enable us to build priors to speed up future learning. Currently, no such work exists in this space. With this in mind, we explore new connections between SVRG and Bayes and assess the potential of SVRG-style ideas for non-traditional deep learning.

## 2.1 Variational Bayes (VB) and Bayesian Learning Rule (BLR)

Throughout, we will use the variational-Bayesian (VB) generalization of empirical-risk minimization (ERM) which optimizes over a tractable set of distributions $q(\boldsymbol{\theta}) \in \mathcal{Q}$ instead of a point estimate $\boldsymbol{\theta} \in \Theta$. This formulation is crucial for us to connect and extend SVRG to Bayes. Below we show an ERM problem on the left and its VB formulation on the right,

$$\boldsymbol{\theta}_* = \arg\min_{\boldsymbol{\theta} \in \Theta} \sum_{i=0}^{N} \ell_i(\boldsymbol{\theta}) \qquad \text{vs} \qquad q_* = \arg\min_{q \in \mathcal{Q}} \sum_{i=1}^{N} \mathbb{E}_q[\ell_i] + \mathbb{D}_{\text{KL}}[q \,\|\, p_0]. \qquad (3)$$

The ERM above contains a regularizer, denoted by $\ell_0$, which is also used to define a valid prior distribution $p_0 \propto \exp(-\ell_0)$ that is added as a Kullback-Leibler (KL) divergence penalty. If $\ell_i$ is a proper likelihood, $q_*$ produces a tractable approximation to the posterior $p_* \propto p_0 \prod_i \exp(-\ell_i)$.

Often we restrict $\mathcal{Q}$ to be exponential-family (EF) distributions, for instance, Gaussians. An EF takes the following log-linear form with respect to a sufficient statistic, denoted by $\mathbf{T}(\boldsymbol{\theta})$, as shown below with an example of an isotropic Gaussian where $\mathbf{T}(\boldsymbol{\theta}) = \boldsymbol{\theta}$,

$$q(\boldsymbol{\theta}) \propto h(\boldsymbol{\theta}) \exp\left(\boldsymbol{\lambda}^\top \mathbf{T}(\boldsymbol{\theta})\right), \qquad \mathcal{N}(\boldsymbol{\theta}|\mathbf{m}, \mathbf{I}) \propto e^{-\frac{1}{2}\boldsymbol{\theta}^\top \boldsymbol{\theta}} \exp\left(\mathbf{m}^\top \boldsymbol{\theta}\right). \qquad (4)$$

The distribution is parameterized by the natural parameter $\boldsymbol{\lambda}$ and uses a base measure $h(\boldsymbol{\theta})$. For an isotropic Gaussian, $\boldsymbol{\lambda} = \mathbf{m}$ and $h(\boldsymbol{\theta}) = \exp(-\frac{1}{2}\boldsymbol{\theta}^\top \boldsymbol{\theta})$. Throughout, we will use the natural parameterization $\boldsymbol{\lambda}$ because updates over it naturally generalize those used in ERM for $\boldsymbol{\theta}$.

Natural gradients are convenient for optimizing Eq. 3, even though for ERM they can be computationally expensive. The convenience is that by using the expectation parameter $\boldsymbol{\mu} = \mathbb{E}_q[\mathbf{T}(\boldsymbol{\theta})]$ we can easily compute the natural gradient of $\mathcal{L}_i(\boldsymbol{\lambda}) = \mathbb{E}_q[\ell_i]$ while avoid computing the Fisher $\mathbf{F}(\boldsymbol{\lambda})$,

$$\widetilde{\nabla}\mathcal{L}_i = \mathbf{F}(\boldsymbol{\lambda})^{-1}\nabla\mathcal{L}_i = \nabla_{\boldsymbol{\mu}}\mathcal{L}_i. \qquad (5)$$

The natural gradient of the KL term can also be simplified. Thus, the natural-gradient descent (NGD) update takes a much simpler form which resembles Bayes' rule. This is called the Bayesian Learning Rule (BLR) (Khan & Rue, 2023) and is shown below in two equivalent forms: using the NGD update (shown on the left) and in its Bayes' form (shown on the right),

$$\boldsymbol{\lambda} \leftarrow (1-\eta)\boldsymbol{\lambda} - \eta \sum_{i=0}^{N} \widetilde{\nabla}\mathcal{L}_i(\boldsymbol{\lambda}) \qquad \Leftrightarrow \qquad q \leftarrow q^{1-\eta} \prod_{i=0}^{N} \exp\left(-\eta\hat{\ell}_i\right). \tag{6}$$

The NGD update is simply a moving average of the natural gradients. The Bayes' form is obtained by substituting $\boldsymbol{\lambda}$ obtained with an NGD step into the EF form of Eq. 4 and by defining the *site* functions $\hat{\ell}_i(\boldsymbol{\theta}) = \widetilde{\nabla}\mathcal{L}_i(\boldsymbol{\lambda})^\top \mathbf{T}(\boldsymbol{\theta})$ of the losses $\ell_i$. The form is equivalent to Bayes' rule with prior $q^{1-\eta}$ and likelihood $\exp(-\eta\hat{\ell}_i)$.

The BLR is attractive not only because it closely mirrors Bayes' rule, but also because it subsumes many ERM algorithms (Khan & Rue, 2023). For example, SGD (Eq. 1) can be derived with isotropic Gaussians $q$ (Eq. 4). Since $\boldsymbol{\lambda} = \mathbf{m}$, $\boldsymbol{\mu} = \mathbb{E}_q[\boldsymbol{\theta}] = \mathbf{m}$, and $\nabla \log h(\boldsymbol{\theta}) = -\boldsymbol{\theta}$, we can write Eq. 6 as

$$\mathbf{m} \leftarrow \mathbf{m} - \eta \sum_{i=0}^{N} \nabla\mathcal{L}_i(\mathbf{m}) \qquad \Leftrightarrow \qquad q \leftarrow q^{1-\eta} \prod_{i=0}^{N} \exp\left(-\eta\nabla\mathcal{L}_i(\mathbf{m})^\top \boldsymbol{\theta}\right). \tag{7}$$

To derive SGD, we use the delta method to approximate $\nabla\mathcal{L}_i(\mathbf{m}) \approx \nabla\ell_i(\mathbf{m})$ and write the stochastic version by redefining $\ell_i + \ell_0/N$. We will use this same strategy to connect gradient correction in SVRG to natural-gradient correction in the BLR. When using $q = \mathcal{N}(\boldsymbol{\theta}|\mathbf{m}, \boldsymbol{\Sigma})$, we obtain the following Newton-like update, called Variational Online Newton (VON), where the pre-conditioner is the precision matrix $\mathbf{S} = \boldsymbol{\Sigma}^{-1}$,

$$\mathbf{m} \leftarrow \mathbf{m} - \eta\mathbf{S}^{-1} \sum_{i=0}^{N} \nabla\mathcal{L}_i(\mathbf{m}), \qquad \text{where} \qquad \mathbf{S} \leftarrow (1-\eta)\mathbf{S} + \eta \sum_{i=0}^{N} \nabla^2\mathcal{L}_i(\mathbf{m}) \tag{8}$$

An improved version of VON is proposed by Lin et al. (2020) and used in Shen et al. (2024) to train large deep networks (such as GPT-2) and obtain competitive results to the Adam optimizer. We will derive a variant of this algorithm later as an extension of SVRG.

## 3 SVRG AND BEYOND VIA POSTERIOR CORRECTION

We will now present a Bayesian approach that generalizes SVRG and enables us to derive new extensions to improve variational deep learning. The approach uses Posterior Correction (PC) (Khan, 2025) which unifies various knowledge transfer tasks, such as, continual learning, federated learning, and model merging. We will now show that it can also be used to recover SVRG.

### 3.1 POSTERIOR CORRECTION

Posterior correction aims to speed-up training by reusing previously computed posteriors, for example, in the form of old checkpoints Khan (2025, Sec. 4.2). We denote the checkpoint by natural parameters $\boldsymbol{\lambda}_{\text{out}}$ and compute the posterior via natural gradients at $\boldsymbol{\lambda}_{\text{out}}$ over the whole dataset:

$$\hat{q}_{\text{out}} \leftarrow \prod_{i=0}^{N} \exp\left(-\hat{\ell}_{i|\text{out}}\right) \quad \text{where } \hat{\ell}_{i|\text{out}}(\boldsymbol{\theta}) = \widetilde{\nabla}\mathcal{L}_i(\boldsymbol{\lambda}_{\text{out}})^\top \mathbf{T}(\boldsymbol{\theta}). \tag{9}$$

We can use this to 'correct' the future BLR updates by simply multiplying and dividing by $q_{\text{out}}^{\eta}$ in the RHS of the Bayes' form shown in Eq. 6, with added parts highlighted in red,

$$q \leftarrow q^{1-\eta}\hat{q}_{\text{out}}^{\eta} \prod_{i=0}^{N} \exp\left(-\eta\left[\hat{\ell}_i - \hat{\ell}_{i|\text{out}}\right]\right). \tag{10}$$

The update remains unchanged because we have simply multiplied it by $\mathbf{1}$, but the new form can be seen as Bayes' rule on a modified model where the prior includes contributions from $q_{\text{out}}$ and the likelihood is 'corrected'. Khan (2025) argue that quick adaptation is possible by reducing the correction and show that many existing schemes perform such corrections. Here, we will use posterior correction to generalize SVRG.

---

**Algorithm 3** VSGD-PC: Variational SGD with posterior correction

---

**Initialize:** Number of inner steps $m$, learning rates $\eta$

1: Initialize $\mathbf{m}_{\text{in}}$.
2: **while** not converged **do**
3: $\quad \mathbf{g}_{\text{out}} \leftarrow \sum_{i=1}^{N} \nabla \ell_i(\boldsymbol{\theta}_{\text{in}})$ where $\boldsymbol{\theta}_{\text{in}} = \mathbf{m}_{\text{in}} + \boldsymbol{\epsilon}$ with $\boldsymbol{\epsilon} \sim \mathcal{N}(0, \mathbf{I})$
4: $\quad \boldsymbol{\theta}_{\text{out}} \leftarrow \boldsymbol{\theta}_{\text{in}}$
5: $\quad$ **for** $t = 1, 2, \ldots, m$ **do**
6: $\quad\quad$ Randomly pick $i \in \{1, 2, \ldots, N\}$
7: $\quad\quad \mathbf{g}_{\text{in}} \leftarrow \nabla \ell_i(\boldsymbol{\theta}_{\text{in}}) - \nabla \ell_i(\boldsymbol{\theta}_{\text{out}}) + \frac{1}{N} \mathbf{g}_{\text{out}}$ where $\boldsymbol{\theta}_{\text{in}} = \mathbf{m}_{\text{in}} + \boldsymbol{\epsilon}$ with $\boldsymbol{\epsilon} \sim \mathcal{N}(0, \mathbf{I})$
8: $\quad\quad \mathbf{m}_{\text{in}} \leftarrow \mathbf{m}_{\text{in}} - \eta \mathbf{g}_{\text{in}}$
9: $\quad$ **end for**
10: **end while**

---

### 3.2 Posterior Correction Generalizes SVRG

To mirror the inner loop Eq. 2 of SVRG, we write a stochastic version of Eq. 10 where one example is sampled. Analogously to Eq. 2, we denote the inner loop iterate as $q_{\text{in}}$. We absorb the regularizer in $\ell_i \leftarrow (\ell_i + \ell_0/N)$, sample a random example $i$ and weight the correction term by $N$ to get

$$q_{\text{in}} \leftarrow q_{\text{in}}^{1-\eta} \hat{q}_{\text{out}}^{\eta} \exp\left(-\eta N \left[\hat{\ell}_{i|\text{in}} - \hat{\ell}_{i|\text{out}}\right]\right). \tag{11}$$

When written in terms of $\boldsymbol{\lambda}$, the update takes an identical form to Eq. 2, as shown below:

$$\boldsymbol{\lambda}_{\text{in}} \leftarrow (1 - \eta)\boldsymbol{\lambda}_{\text{in}} + \eta \hat{\boldsymbol{\lambda}}_{\text{out}} - \eta N \left[\widetilde{\nabla}\mathcal{L}_i(\boldsymbol{\lambda}_{\text{in}}) - \widetilde{\nabla}\mathcal{L}_i(\boldsymbol{\lambda}_{\text{out}})\right] \tag{12}$$

$$\implies \boldsymbol{\lambda}_{\text{in}} \leftarrow (1 - \eta)\boldsymbol{\lambda}_{\text{in}} - \eta N \left[\widetilde{\nabla}\mathcal{L}_i(\boldsymbol{\lambda}_{\text{in}}) - \widetilde{\nabla}\mathcal{L}_i(\boldsymbol{\lambda}_{\text{out}}) + \frac{1}{N}\sum_{j=1}^{N}\widetilde{\nabla}\mathcal{L}_j(\boldsymbol{\lambda}_{\text{out}})\right]. \tag{13}$$

The second update is obtained by using Eq. 9 which shows that $\hat{\boldsymbol{\lambda}}_{\text{out}} = \sum_i \widetilde{\nabla}\mathcal{L}_i(\boldsymbol{\lambda}_{\text{out}})$. The update is strikingly similar to Eq. 2 with all instances of gradients $\nabla \ell_i$ replaced by natural gradients $\widetilde{\nabla}\mathcal{L}_i$. Alg. 2 shows this algorithm, where we denote $\hat{\boldsymbol{\lambda}}_{\text{out}}$ by $\widetilde{\mathbf{g}}_{\text{out}}$ and use the BLR in the inner-loop update.

We will now show our main result that SVRG is a special case of Alg. 2, when we use PC over the isotropic-Gaussian family, that is, we set $\mathcal{Q}$ to be a set of $q_{\text{in}} = \mathcal{N}(\boldsymbol{\theta}|\mathbf{m}_{\text{in}}, \mathbf{I})$.

**Theorem 1** *For isotropic-Gaussian family, Eq. 13 reduces to the following update of the mean $\mathbf{m}_{in}$,*

$$\mathbf{m}_{in} \leftarrow \mathbf{m}_{in} - \eta N \left[\mathbb{E}_{q_{in}}[\nabla \ell_i] - \mathbb{E}_{q_{out}}[\nabla \ell_i] + \frac{1}{N}\sum_{j=1}^{N}\mathbb{E}_{q_{out}}[\nabla \ell_j]\right]. \tag{14}$$

The proof follows similarly to Eq. 7 by plugging-in the definitions of $\boldsymbol{\lambda}, \boldsymbol{\mu}$, and $h(\boldsymbol{\theta})$, and using Bonnet's theorem (Bonnet, 1964) which states that $\nabla_{\mathbf{m}}\mathcal{L}_i = \mathbb{E}_q[\nabla \ell_i]$. That is, the natural gradients can be computed by computing the expectation of $\ell_i(\boldsymbol{\theta})$ at sample $\boldsymbol{\theta} \sim q$, which is convenient for implementation. An algorithm to implement the update is given in Alg. 3, which we call VSGD-PC and where for simplicity we use one Monte Carlo (MC) sample. VSGD-PC and using SVRG with SGD have almost the same cost, except for the additional parameter sampling which is fast to compute. The algorithm is almost identical to SVRG (Alg. 1) with differences highlighted in red but can exactly match SVRG as shown below.

**Theorem 2** *SVRG (Alg. 1) is equivalent to VSGD-PC (Alg. 3) where we set $\boldsymbol{\epsilon} = 0$ in line 3 and 7.*

Setting $\boldsymbol{\epsilon} = 0$ is equivalent to applying the delta method: $\mathbb{E}_{q_{\mathbf{m}}}[\nabla \ell_i] \approx \nabla \ell_i(\mathbf{m})$ which we also use in Eq. 7 to recover SGD from the BLR. We will show in the next section that, by using other EF forms, we can derive new extensions that go beyond SVRG. We note that this is the first result of its kind connecting SVRG and Bayes. Previous works have applied SVRG to improve ELBO optimization (Zhang et al., 2018), but are merely using SVRG and unable to recover or extend it.

## 3.3 Beyond SVRG: New Extensions Derived using Posterior Correction

In this section, we derive new extensions to boost variational training of deep networks.

**A Newton-like Variant:** The original SVRG and many of its variants focus on stabilizing the gradient and little work has been done on methods that do the same for the Hessian. We derive such variants by using more flexible Gaussian forms than isotropic Gaussians, for example, the multivariate Gaussian $q = \mathcal{N}(\boldsymbol{\theta}|\mathbf{m}, \mathbf{S}^{-1})$, where the precision matrix $\mathbf{S}$ is also estimated. Then, Alg. 2 reduces to a posterior correction version of the Variational-Online-Newton (VON) algorithm shown in Eq. 8. The update shown below employs a stochastic variance reduction for the Hessian.

**Theorem 3** *For Gaussian $q_{in} = \mathcal{N}(\mathbf{m}_{in}, \mathbf{S}_{in}^{-1})$, Eq. 13 reduces to a Newton-like update,*

$$\mathbf{m}_{in} = \mathbf{m}_{in} - \eta N \mathbf{S}_{in}^{-1} \left[ \mathbb{E}_{q_{in}}[\nabla \ell_i] - \mathbb{E}_{q_{out}}[\nabla \ell_i] + \frac{1}{N} \sum_{j=1}^{N} \mathbb{E}_{q_{out}}[\nabla \ell_j] + \mathbf{H}_{out\setminus i}(\mathbf{m}_{in} - \mathbf{m}_{out}) \right] \quad (15)$$

*where we use a Stochastic Variance-Reduced Hessian (SVRH) estimate as the pre-conditioner*

$$\mathbf{S}_{in} \leftarrow (1 - \eta)\mathbf{S}_{in} + \eta N \left[ \mathbb{E}_{q_{in}}[\nabla^2 \ell_i] + \bar{\mathbf{H}}_{out\setminus i} \right]. \quad (16)$$

*Here, $\bar{\mathbf{H}}_{out\setminus i} = \frac{1}{N} \sum_{j=1}^{N} \mathbb{E}_{q_{out}}[\nabla^2 \ell_j] - \mathbb{E}_{q_{out}}[\nabla^2 \ell_i]$ is the full-batch expected Hessian without $\ell_i$.*

The derivation uses the definition of natural parameter and gradient in Eq. 13 and is given in App. A. An algorithm is in Alg. 5 which we name VON-PC, as a posterior correction version of VON.

We are not aware of any other Newton variant that implements similar Hessian corrections as shown above. Most works on Newton steps only use corrections for the gradient and never for the Hessian (Derezinski, 2023; Sadiev et al., 2024; Garg et al., 2024; Sun et al., 2025). One surprising connection is that the term $\mathbf{H}_{out\setminus i}(\mathbf{m}_{in} - \mathbf{m}_{out})$ in Eq. 15 is also used in Chayti et al. (2024, Eqs. 11–12) but is derived differently via cubic-Newton. The method has rigorous guarantees and its relation to our Bayesian approach remains an interesting case to study. The term can also be viewed as forcing the inner iteration to stay close to the most recent outer iteration.

The PC method can be applied to any EF distribution and therefore yields novel extensions that go way beyond SVRG, for example, for binary neural networks (Meng et al., 2020) to yield SVRG-style updates for the Straight-Through Estimator (Bengio et al., 2013) via Bernoulli distributions. We omit the derivation because the procedure is similar to the ones we presented here. The PC method can yield novel extensions of SVRG by exploiting flexible EF distributions.

**An Adam-like variant:** A cheaper Adam-like variant is obtained by using diagonal covariance, for example, $q = \mathcal{N}(\boldsymbol{\theta}|\mathbf{m}, \text{diag}(\mathbf{s})^{-1})$. This implements posterior correction over IVON (Shen et al., 2024). A detailed derivation is in App. B and a pseudo-code of IVON-PCM is in Alg. 4, where we highlight additional computation on top of IVON. Similarly to IVON, IVON-PCM uses several practical tricks, for instance, weight decay, mini-batching, temperature, momentum, and debiasing. If momentum is switched off, i.e. $\rho_1 = \rho_2 = 1$, we simply refer to the metho as IVON-PC.

**Handling Mega-Batches and Weighted Corrections:** The version shown in Alg. 4 also avoids expensive full-batch computations. Instead, it uses *mega*-batches (Defazio & Bottou, 2019) and builds online estimates of gradients and Hessians (line 3-4). Mega-batches can be tens of times the size of the inner loop mini-batches and can be used to slowly build an estimate of full-batch gradients and Hessians via an estimate of the site function, for example, for isotropic Gaussians:

$$\hat{q}_{\text{out}} \leftarrow \exp\left(-\boldsymbol{\theta}^\top \mathbf{g}_{\text{out}}\right), \quad \text{where } \mathbf{g}_{\text{out}} \leftarrow \rho_1 \mathbf{g}_{\text{out}} + (1 - \rho_1) \sum_{i \in \mathcal{M}} \mathbb{E}_{q_{\text{out}}}[\nabla \ell_i]. \quad (17)$$

A similar approach for full-Gaussians is in App. B.1. Posteriors constucted using mega-batches can be imperfect and downweighed by modifying the PC update of Eq. 11 using $\alpha < 1$,

$$q_{\text{in}} \leftarrow q_{\text{in}}^{1-\eta} \hat{q}_{\text{out}}^{\eta\alpha} \exp\left(-\eta N \left[\hat{\ell}_{i|\text{in}} - \alpha \hat{\ell}_{i|\text{out}}\right]\right), \quad (18)$$

For $\alpha = 0$, the update reduces to standard BLR, while for $\alpha = 1$ we use perfect corrections which are good for full-batch $\hat{q}_{\text{out}}$. When using mega-batches, a smaller value could be used and tuned.

---

**Algorithm 4** IVON-PCM: IVON with Posterior Correction and Momentum. IVON-PC is obtained by removing momentum (differences to IVON highlighted in red)

---

**Require:** Learning rates $\{\eta_t\}$, $\beta_1 \in [0, 1)$, $\beta_2 \in [0, 1)$, $\delta > 0$, $\kappa > 0$, $h_0 > 0$, clip radius $\xi > 0$, mini-batch size $B$, mega-batch size $M$, outer loop learning rate $\rho_1, \rho_2$, refresh rate $\alpha$.

1: Initialize: $\mathbf{m}_{\text{in}} \leftarrow$ (NN weight init), $\mathbf{h}_{\text{in}} \leftarrow h_0$, $\boldsymbol{\sigma}_{\text{in}} \leftarrow 1/\sqrt{\kappa(\mathbf{h}_{\text{in}} + \delta)}$, $\mathbf{g} \leftarrow 0$

2: **while** not converged **do**

3:    $\widehat{\mathbf{g}}_{\text{out}} \leftarrow \frac{1}{M} \sum_{i \in \mathcal{M}} \nabla \ell_i(\boldsymbol{\theta}_{\text{in}})$    where we sample a mega-batch $\mathcal{M}$ and $\boldsymbol{\theta}_{\text{in}} \sim \mathcal{N}(\mathbf{m}_{\text{in}}, \boldsymbol{\sigma}_{\text{in}}^2)$

4:    $\mathbf{g}_{\text{out}} \leftarrow \rho_1 \mathbf{g}_{\text{out}} + (1 - \rho_1)\widehat{\mathbf{g}}_{\text{out}}$,   and   $\mathbf{h}_{\text{out}} \leftarrow \rho_2 \mathbf{h}_{\text{out}} + (1 - \rho_2)\widehat{\mathbf{g}}_{\text{out}}(\boldsymbol{\theta}_{\text{in}} - \mathbf{m}_{\text{in}})/\boldsymbol{\sigma}_{\text{in}}^2$

5:    $\mathbf{m}_{\text{out}} \leftarrow \mathbf{m}_{\text{in}}, \; \boldsymbol{\sigma}_{\text{out}} \leftarrow \boldsymbol{\sigma}_{\text{in}}$

6:    **for** $t = 1, 2, \ldots, m$ **do**

7:       Sample a mini-batch $\mathcal{B}$, $\boldsymbol{\theta}_{\text{in}} \sim \mathcal{N}(\mathbf{m}_{\text{in}}, \boldsymbol{\sigma}_{\text{in}}^2)$, $\boldsymbol{\theta}_{\text{out}} \sim \mathcal{N}(\mathbf{m}_{\text{out}}, \boldsymbol{\sigma}_{\text{out}}^2)$

8:       $\widehat{\mathbf{g}}_{\text{in}} \leftarrow \frac{1}{B} \sum_{i \in \mathcal{B}} \nabla \ell_i(\boldsymbol{\theta}_{\text{in}})$    and   $\widehat{\mathbf{h}}_{\text{in}} \leftarrow \widehat{\mathbf{g}}_{\text{in}}(\boldsymbol{\theta}_{\text{in}} - \mathbf{m}_{\text{in}})/\boldsymbol{\sigma}_{\text{in}}^2$

9:       $\widehat{\mathbf{g}}_{\text{out}} \leftarrow \frac{1}{B} \sum_{i \in \mathcal{B}} \nabla \ell_i(\boldsymbol{\theta}_{\text{out}})$    and   $\widehat{\mathbf{h}}_{\text{out}} \leftarrow \hat{\mathbf{g}}_{\text{out}}(\boldsymbol{\theta}_{\text{out}} - \mathbf{m}_{\text{out}})/\boldsymbol{\sigma}_{\text{out}}^2$

10:     $\hat{\mathbf{g}} \leftarrow \hat{\mathbf{g}}_{\text{in}} - \alpha(\hat{\mathbf{g}}_{\text{out}} - \mathbf{g}_{\text{out}})$    and   $\widehat{\mathbf{h}} \leftarrow \widehat{\mathbf{h}}_{\text{in}} - \alpha(\widehat{\mathbf{h}}_{\text{out}} - \mathbf{h}_{\text{out}})$

11:     $\mathbf{g} \leftarrow \beta_1 \mathbf{g} + (1 - \beta_1)\hat{\mathbf{g}}$

12:     $\mathbf{h} \leftarrow \beta_2 \mathbf{h} + (1 - \beta_2)\widehat{\mathbf{h}} + \frac{1}{2}(1 - \beta_2)^2(\mathbf{h} - \widehat{\mathbf{h}})^2/(\mathbf{h} + \delta)$

13:     $\bar{\mathbf{g}} \leftarrow \mathbf{g}/(1 - \beta_1^t)$

14:     $\bar{\mathbf{g}} \leftarrow (\bar{\mathbf{g}} + \delta \mathbf{m}_{\text{in}} + \alpha(\mathbf{h}_{\text{out}} - \widehat{\mathbf{h}}_{\text{out}})(\mathbf{m}_{\text{in}} - \mathbf{m}_{\text{out}}))/(\mathbf{h} + \delta)$

15:     $\mathbf{m}_{\text{in}} \leftarrow \mathbf{m}_{\text{in}} - \eta_t \, \text{clip}(\bar{\mathbf{g}}, \xi)$

16:     $\boldsymbol{\sigma}_{\text{in}} \leftarrow 1/\sqrt{\kappa(\mathbf{h} + \delta)}$

17:   **end for**

18: **end while**

19: **return** $\mathbf{m}, \boldsymbol{\sigma}$

---

When applied to isotropic Gaussians, this reduces to $\alpha$-SVRG (Yin et al., 2025), though their motivation to use $\alpha$ does not stem from the use of mega-batches (they appear to use full batches). Their motivation is to reduce variance early on via scheduling $\alpha$, starting with a high value. From a Bayesian perspective, the $\hat{q}_{\text{out}}$ estimates are expected to be less useful early on. Thus, it makes more sense to use a 'burn-in' period with $\alpha = 0$ and then turn on $\alpha$. In our experiments, we use a constant $\alpha$, which seems to work better for deep learning.

### 3.4 Computational & Memory Requirements

The computation and memory overheads of IVON-PCM are similar to the use of Adam to implement alpha-SVRG. Adam uses an accumulation of squared gradients, while here reparameterization trick is used to compute a Hessian estimate (line 4, 8, 9). Unlike alpha-SVRG, IVON-PCM use an additional Hessian correction as well, but this does not add a significant cost because the Hessian is already computed. We just need to store an additional $\mathbf{h}_{\text{out}}$ in the outer loop (in line 9), as well as $\boldsymbol{\sigma}_{\text{out}}$ (line 5). Similarly to VSGD-PC, sampling is added in line 3 and 7. All of these costs are not significant. Just like in SVRG and alpha-SVRG, the major overhead is the mega-batch computation and the use of two gradient (line 10). An extra Hessian correction is used in line 10 as well.

## 4 Experiments & Results

### 4.1 Logistic Regression

We first illustrate our new method IVON-PC on simple convex logistic regression problems of varying dimensionality and number of data examples in Fig. 3. We show additional results on CIFAR-10 logistic regression in App. C. We compare IVON-PC against IVON, SVRG, and SGD. For all experiments, we used a large constant learning rate and a batch size of 5. For the IVON methods, we set $\rho_1 = \rho_2 = 1$ (i.e. no momentum) and downweight the extra terms in line 17 of Alg. 4 by 0.01. The effects of varrying $\rho$ and the additional term are explored in our ablation experiments in App. C.

Consistent with the original work by Johnson & Zhang (2013), our experiments show that SVRG drastically boosts the performance of SGD once the first outer loop—illustrated by the gray bars,

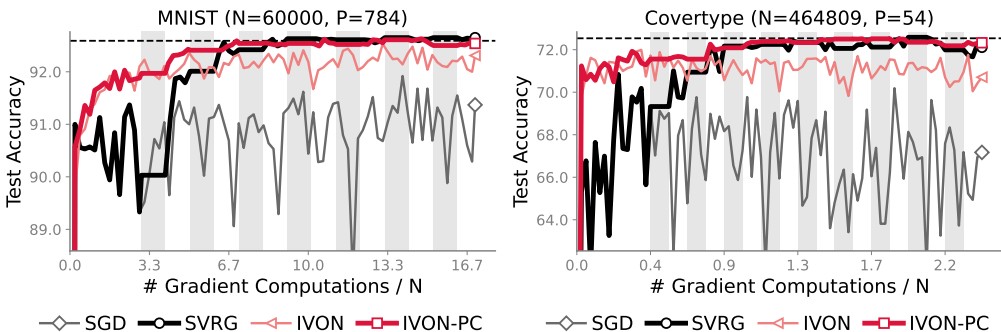

Figure 3: IVON-PC significantly boosts the convergence speed of IVON and performs much better than SVRG, here on three convex logistic regression problems of varying dimension and size. The horizontal dashed line indicates the performance at the minimum, the gray bars indicate outer gradient computations used in SVRG and IVON-PC.

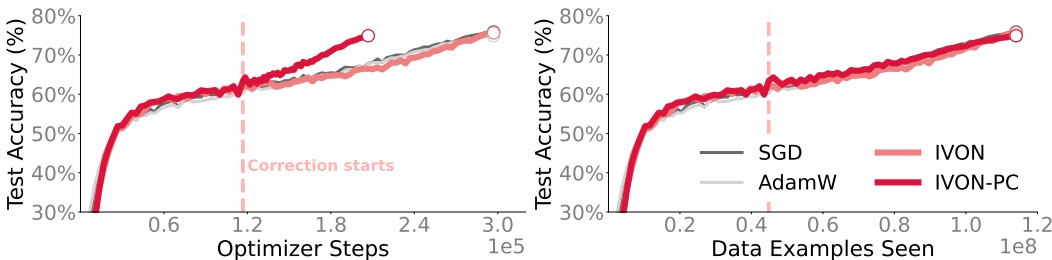

Figure 4: Performance on ImageNet for ResNet-50. When comparing by the number of optimization steps (left) IVON-PC gives clear improvements but not in terms of data examples seen (right). On the left, we zoom in on the final stage of training when correction is added.

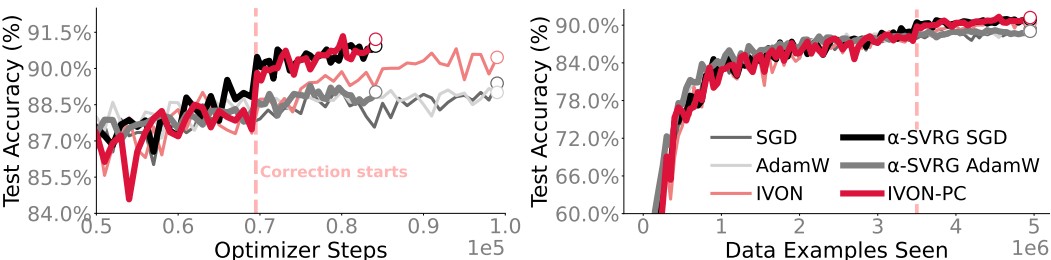

Figure 5: Comparison to $\alpha$-SVRG on CIFAR-10 for ResNet-20. IVON-PC and $\alpha$-SVRG with SGD give improvements, also when counting the number of data examples seen (right). At the end of training, IVON-PC performs best out of all methods.

which indicate the large outer batch size—is hit. Similarly, IVON-PC boosts the performance of IVON. Since IVON already outperforms SGD and is comparable to SVRG due to its inherent preconditioning and momentum, the further boost provided by IVON-PC makes it perform best.

## 4.2 IMAGE CLASSIFICATION WITH RESNETS

We first evaluate our proposed IVON-PC method on training a ResNet-50 on ImageNet. Fig. 4 shows the results. We can see that IVON-PC reaches a comparable accuracy to SGD, IVON and AdamW in much fewer optimization steps. Factoring in the outer loop gradient calculation, all methods perform similarly well. Variance reduction methods such as SVRG have classically struggled on training ResNets on ImageNet (Defazio & Bottou, 2019). We observe a similar effect but performance is greatly improved when just counting optimization steps as the recent work by Yin et al. (2025).

Table 1: IVON-PC can improve finetuning. All methods use the same number of steps. We indicate with "+" improvements over the IVON baseline.

|  | ViT-B/32 | | | | Qwen2.5-0.5B-it | | Llama-3.1-8B |
|  | Cars | DTD | GTSRB | RESISC45 | XSUM (R-1, R-L) | | GSM8k |
|---|---|---|---|---|---|---|---|
| IVON | 79.5 | 72.9 | **99.9** | 95.2 | 48.7 | 23.6 | 30.4 |
| + PC | **80.0**$_{+0.5}$ | **73.4**$_{+0.5}$ | **99.9**$_{+0.0}$ | **96.1**$_{+0.9}$ | **49.6**$_{+0.9}$ | **23.8**$_{+0.2}$ | **30.6**$_{+0.2}$ |

In a second experiment on a smaller ResNet-20 in Fig. 5, we do not anneal the learning rate to zero but to a quarter of the starting learning rate. Then, the variance is not completely removed through learning rate annealing and both $\alpha$-SVRG with SGD and IVON-PC bring improvements in accuracy over their respective baseline methods, with IVON-PC having the advantage of estimating a variational posterior distribution. All details for the two experiments are in App. D.1.

## 4.3 LANGUAGE MODEL PRETRAINING

Here, we present results when pretraining a 125M parameter GPT-2 model from scratch on 50B tokens from the OpenWebText dataset[1]. We follow the set-up from Shen et al. (2024) and train each model for 100,000 steps using AdamW, IVON, and IVON-PCM. For IVON-PCM we use two different configurations: One refreshes a megabatch with size 10 times the minibatch size after 10 inner step starting correction after 50,000 steps and another where we start correction at varying points, namely, after 30,000, 50,000, and 70,000 steps with a megabatch factor and refresh rate of 20 as opposed to 10. Results are shown in Fig. 1b and Fig. 1c, respectively. We find that IVON-PCM provides improvements in terms of validation perplexity in both settings. Interestingly, correction can be started at varying intervals and benefits, especially a direct jump downwards of the validation perplexity, are maintained. Details are found in App. D.2.

## 4.4 CONTINUAL PRETRAINING

Next, we present results on continually pretraining the GPT-125M model from Shen et al. (2024) on 1B tokens from Fineweb-edu (Penedo et al., 2024). We compare AdamW and alpha-SVRG with AdamW against IVON, IVON-PC, and IVON-PCM. Both alpha-SVRG and the IVON-PC runs use $\alpha = 0.7$, 40 steps for the inner loop, and 1,000 warmup steps without correction. Results are in Fig. 6. Adding correction both with $\alpha$-SVRG and IVON-PC helps but the gap between IVON, which struggles with larger learning rates, and IVON-PC is larger. Adding Momentum to IVON-PC improves results beyond those of $\alpha$-SVRG and IVON-PCM converges to a better validation perplexity after the same number of steps. When comparing time, $\alpha$-SVRG does not improve over AdamW but IVON-PCM improves over IVON. Details can be found in App. D.3.

## 4.5 FINETUNING

Here, we use IVON-PC for finetuning different Transformers, namely ViT-B/32 (Dosovitskiy et al., 2021), Qwen2.5-0.5B-Instruct (Yang et al., 2025), and Llama-3.1-8B (Dubey et al., 2024) on image classification, XSUM, and GSM8k, respectively. For the ViT models we finetune only the image coder, for Qwen we finetune the entire model, and for Llama-3.1 we use LoRA finetuning (Hu et al., 2022). Results are shown in Table 1 and show that IVON-PC can improve final performance over IVON when the models are trained with the same number of optimization steps. Details and hyperparameters used for these experiments can be found in App. D.4.

## 4.6 ABLATIONS

**Influence of $\alpha$:** We study the influence of $\alpha$ for $\alpha$-SVRG and IVON-PC on wikitext103 (Merity et al., 2017) by training a GPT-2-based model with 33M parameters from scratch for one epoch with batch size of 64 and 5000 warmup steps without correction. Results are shown in Fig. 6 (left): for both methods using correction improves performance over various values of $\alpha$. Both times the

---

[1]https://huggingface.co/datasets/Skylion007/openwebtext

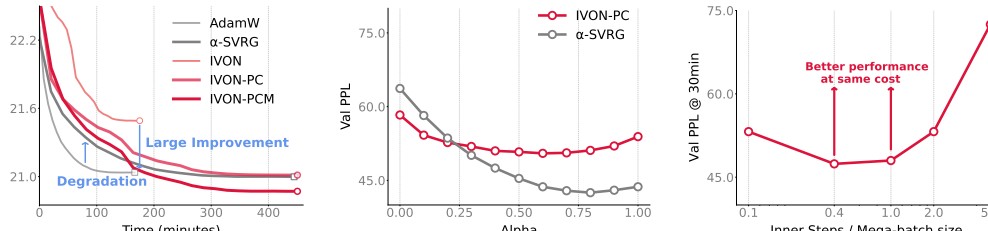

Figure 6: (left) IVON-PC and IVON-PCM also boost IVON's performance when continually pre-training a GPT2-125M model on 1B tokens from Fineweb-Edu, here measured in terms of wallclock time. This is unlike AdamW that does not benefit from SVRG when using $\alpha$-SVRG. (center) Tuning $\alpha$ has a large effect on performance on wikitext103 with a 33M Transformer trained from scratch. (right) In the same setting, increasing the number of refreshes can help but at some point becomes slow, with better performance at the same time budget obtained by fewer refreshes.

curve has a u-shape indicating that biasing too much towards large batches can harm performance. We provide details for these and the following analysis of the inner loop size in App. D.2.

**Influence of Inner Loop Size** Here, we fix the megabatch size to $50 \cdot 64$ in the same setting as above and vary the number of refreshes. Fig. 6 (center) shows that refreshing more often in general helps performance but when refreshing too often performance can get worse at the same compute budget.

## 5 CONCLUSION

In this paper, we present surprising new connections between two seemingly unrelated ideas: SVRG and posterior correction (PC). The connections allow us to derive new extensions of SVRG which can boost variational training of deep networks. We hope that the non-traditional settings considered in this paper encourage researchers who wish to see SVRG successfully applied to deep learning. Our work attempts to offer a new perspective on variance reduction as knowledge transfer, giving SVRG-style ideas a fresh restart. In the future, we hope to apply these ideas to other non-traditional problems where reusing past knowledge is crucial for reducing cost.

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

# A  DERIVATION OF THE NEWTON-LIKE SVRG EXTENSION

We start by writing $q$ in an exponential-family form which in this case is convenient to write in terms of the precision $\mathbf{S} = \boldsymbol{\Sigma}^{-1}$. This is shown below with sufficient statistics highlighted in red,

$$\mathcal{N}(\boldsymbol{\theta}|\mathbf{m}, \mathbf{S}^{-1}) \propto \exp\left[\mathbf{m}^\top \mathbf{S}\boldsymbol{\theta} + \mathrm{Tr}\left((-\tfrac{1}{2}\mathbf{S})\boldsymbol{\theta}\boldsymbol{\theta}^\top\right)\right]. \tag{19}$$

This is a log-linear form for a sufficient-statistics vector of $\boldsymbol{\theta}$ and $\boldsymbol{\theta}\boldsymbol{\theta}^\top$. We denote the natural parameter $\boldsymbol{\lambda} = (\mathbf{S}\mathbf{m}, -\tfrac{1}{2}\mathbf{S})$ consisting of two elements: a vector and a square matrix. The natural gradient can be written in terms of the gradient and Hessian of $\ell_i$ (Khan & Rue, 2023, Eq. 10-11),

$$\widetilde{\nabla}\mathcal{L}_i(\boldsymbol{\lambda}) = \mathbb{E}_q\left[\left(\nabla\ell_i - \nabla^2\ell_i\mathbf{m}, \ \tfrac{1}{2}\nabla^2\ell_i\right)\right]. \tag{20}$$

This also has two elements, and uses gradient and Hessian computed at samples from $q$.

We will denote the $\boldsymbol{\lambda}_{\mathrm{in}} = (\mathbf{S}_{\mathrm{in}}\mathbf{m}_{\mathrm{in}}, -\tfrac{1}{2}\mathbf{S}_{\mathrm{in}})$ and $\boldsymbol{\lambda}_{\mathrm{out}} = (\mathbf{S}_{\mathrm{out}}\mathbf{m}_{\mathrm{out}}, -\tfrac{1}{2}\mathbf{S}_{\mathrm{out}})$. We first write the update for the second entry of $\boldsymbol{\lambda}_{\mathrm{in}}$ which is $-\tfrac{1}{2}\mathbf{S}_{\mathrm{in}}$,

$$\mathbf{S}_{\mathrm{new}} \leftarrow (1-\eta)\mathbf{S}_{\mathrm{in}} + \eta N\left[\mathbb{E}_{q_{\boldsymbol{\lambda}_{\mathrm{in}}}}[\nabla^2\ell_i] - \mathbb{E}_{q_{\boldsymbol{\lambda}_{\mathrm{out}}}}[\nabla^2\ell_i] + \frac{1}{N}\sum_{j=1}^N \mathbb{E}_{q_{\boldsymbol{\lambda}_{\mathrm{out}}}}[\nabla^2\ell_j]\right]. \tag{21}$$

We denoted the new value as $\mathbf{S}_{\mathrm{new}}$ to differentiate it with the old value. This will be useful to simplify the update for the first entry $\mathbf{S}_{\mathrm{in}}\mathbf{m}_{\mathrm{in}}$ which we show below,

$$\mathbf{S}_{\mathrm{new}}\mathbf{m}_{\mathrm{new}} = (1-\eta)\mathbf{S}_{\mathrm{in}}\mathbf{m}_{\mathrm{in}} - \eta N\left[\mathbb{E}_{q_{\boldsymbol{\lambda}_{\mathrm{in}}}}[\nabla\ell_i - \nabla^2\ell_i\mathbf{m}_{\mathrm{in}}] - \mathbb{E}_{q_{\boldsymbol{\lambda}_{\mathrm{out}}}}[\nabla\ell_i - \nabla^2\ell_i\mathbf{m}_{\mathrm{out}}]\right.$$

$$\left. + \frac{1}{N}\sum_{j=1}^N \mathbb{E}_{q_{\boldsymbol{\lambda}_{\mathrm{out}}}}[\nabla\ell_j - \nabla^2\ell_j\mathbf{m}_{\mathrm{out}}]\right]$$

$$= \left\{\mathbf{S}_{\mathrm{new}} - \eta N\left[\mathbb{E}_{q_{\boldsymbol{\lambda}_{\mathrm{in}}}}[\nabla^2\ell_i] - \mathbb{E}_{q_{\boldsymbol{\lambda}_{\mathrm{out}}}}[\nabla^2\ell_i] + \frac{1}{N}\sum_{j=1}^N \mathbb{E}_{q_{\boldsymbol{\lambda}_{\mathrm{out}}}}[\nabla^2\ell_j]\right]\right\}\mathbf{m}_{\mathrm{in}} +$$

$$- \eta N\left[\mathbb{E}_{q_{\boldsymbol{\lambda}_{\mathrm{in}}}}[\nabla\ell_i - \nabla^2\ell_i\mathbf{m}_{\mathrm{in}}] - \mathbb{E}_{q_{\boldsymbol{\lambda}_{\mathrm{out}}}}[\nabla\ell_i - \nabla^2\ell_i\mathbf{m}_{\mathrm{out}}]\right.$$

$$\left. + \frac{1}{N}\sum_{j=1}^N \mathbb{E}_{q_{\boldsymbol{\lambda}_{\mathrm{out}}}}[\nabla\ell_j - \nabla^2\ell_j\mathbf{m}_{\mathrm{out}}]\right]$$

$$= \mathbf{S}_{\mathrm{new}}\mathbf{m}_{\mathrm{in}} - \eta N\left[\cancel{\mathbb{E}_{q_{\boldsymbol{\lambda}_{\mathrm{in}}}}[\nabla^2\ell_i]\mathbf{m}_{\mathrm{in}}} - \mathbb{E}_{q_{\boldsymbol{\lambda}_{\mathrm{out}}}}[\nabla^2\ell_i]\mathbf{m}_{\mathrm{in}} + \frac{1}{N}\sum_{j=1}^N \mathbb{E}_{q_{\boldsymbol{\lambda}_{\mathrm{out}}}}[\nabla^2\ell_j]\mathbf{m}_{\mathrm{in}}\right.$$

$$\left. + \mathbb{E}_{q_{\boldsymbol{\lambda}_{\mathrm{in}}}}[\nabla\ell_i - \cancel{\nabla^2\ell_i\mathbf{m}_{\mathrm{in}}}] - \mathbb{E}_{q_{\boldsymbol{\lambda}_{\mathrm{out}}}}[\nabla\ell_i - \nabla^2\ell_i\mathbf{m}_{\mathrm{out}}] + \frac{1}{N}\sum_{j=1}^N \mathbb{E}_{q_{\boldsymbol{\lambda}_{\mathrm{out}}}}[\nabla\ell_j - \nabla^2\ell_j\mathbf{m}_{\mathrm{out}}]\right]$$

$$= \mathbf{S}_{\mathrm{new}}\mathbf{m}_{\mathrm{in}} - \eta N\left[\mathbb{E}_{q_{\boldsymbol{\lambda}_{\mathrm{in}}}}[\nabla\ell_i] - \mathbb{E}_{q_{\boldsymbol{\lambda}_{\mathrm{out}}}}[\nabla\ell_i] + \frac{1}{N}\sum_{j=1}^N \mathbb{E}_{q_{\boldsymbol{\lambda}_{\mathrm{out}}}}[\nabla\ell_j] - \mathbb{E}_{q_{\boldsymbol{\lambda}_{\mathrm{out}}}}[\nabla^2\ell_i]\mathbf{m}_{\mathrm{in}}\right.$$

$$\left. + \frac{1}{N}\sum_{j=1}^N \mathbb{E}_{q_{\boldsymbol{\lambda}_{\mathrm{out}}}}[\nabla^2\ell_j]\mathbf{m}_{\mathrm{in}} + \mathbb{E}_{q_{\boldsymbol{\lambda}_{\mathrm{out}}}}[\nabla^2\ell_i\mathbf{m}_{\mathrm{out}}] - \frac{1}{N}\sum_{j=1}^N \mathbb{E}_{q_{\boldsymbol{\lambda}_{\mathrm{out}}}}[\nabla^2\ell_j\mathbf{m}_{\mathrm{out}}]\right]$$

$$= \mathbf{S}_{\mathrm{new}}\mathbf{m}_{\mathrm{in}} - \eta N\left[\mathbb{E}_{q_{\boldsymbol{\lambda}_{\mathrm{in}}}}[\nabla\ell_i] - \mathbb{E}_{q_{\boldsymbol{\lambda}_{\mathrm{out}}}}[\nabla\ell_i] + \frac{1}{N}\sum_{j=1}^N \mathbb{E}_{q_{\boldsymbol{\lambda}_{\mathrm{out}}}}[\nabla\ell_j]\right.$$

$$\left. \left(\frac{1}{N}\sum_{j=1}^N \mathbb{E}_{q_{\boldsymbol{\lambda}_{\mathrm{out}}}}[\nabla^2\ell_j] - \mathbb{E}_{q_{\boldsymbol{\lambda}_{\mathrm{out}}}}[\nabla^2\ell_i]\right)(\mathbf{m}_{\mathrm{in}} - \mathbf{m}_{\mathrm{out}})\right]$$

$$\tag{22}$$

---

**Algorithm 5** VON-PC: Variational Online Newton with Posterior Correction

---

**Initialize:** Number of inner steps $m$, learning rates $\alpha$ and $\beta$

1: Initialize $\mathbf{m}_{\text{in}}, \mathbf{S}_{\text{in}}$
2: **while** not converged **do**
3:      $\mathbf{g}_{\text{out}} \leftarrow \sum_{i=1}^{N} \nabla\ell_i(\boldsymbol{\theta}_{\text{in}})$ where $\boldsymbol{\theta}_{\text{in}} = \mathbf{m}_{\text{in}} + \mathbf{S}_{\text{in}}^{-\frac{1}{2}}\boldsymbol{\epsilon}$ with $\boldsymbol{\epsilon} \sim \mathcal{N}(0, \mathbf{I})$
4:      $\mathbf{H}_{\text{out}} \leftarrow \sum_{i=1}^{N} \nabla^2\ell_i(\boldsymbol{\theta}_{\text{in}})$
5:      $\boldsymbol{\theta}_{\text{out}} \leftarrow \boldsymbol{\theta}_{\text{in}}, \mathbf{m}_{\text{out}} \leftarrow \mathbf{m}_{\text{in}}$
6:      **for** $t = 1, 2, \ldots, m$ **do**
7:          Randomly pick $i \in \{1, 2, \ldots, N\}$
8:          $\mathbf{g}_{\text{in}} \leftarrow \nabla\ell_i(\boldsymbol{\theta}_{\text{in}}) - \nabla\ell_i(\boldsymbol{\theta}_{\text{out}}) + \frac{1}{N}\mathbf{g}_{\text{out}}$ where $\boldsymbol{\theta}_{\text{in}} = \mathbf{m}_{\text{in}} + \mathbf{S}_{\text{in}}^{-\frac{1}{2}}\boldsymbol{\epsilon}$ with $\boldsymbol{\epsilon} \sim \mathcal{N}(0, \mathbf{I})$
9:          $\mathbf{H}_{\text{out}\backslash i} \leftarrow \frac{1}{N}\mathbf{H}_{\text{out}} - \nabla^2\ell_i(\boldsymbol{\theta}_{\text{out}})$
10:        $\mathbf{H}_{\text{in}} \leftarrow \nabla^2\ell_i(\boldsymbol{\theta}_{\text{in}}) + \mathbf{H}_{\text{out}\backslash i}$
11:        $\mathbf{S}_{\text{in}} \leftarrow (1-\beta)\mathbf{S}_{\text{in}} + \beta N \mathbf{H}_{\text{in}}$
12:        $\mathbf{m}_{\text{in}} \leftarrow \mathbf{m}_{\text{in}} - \alpha N \mathbf{S}_{\text{in}}^{-1}\left[\mathbf{g}_{\text{in}} + \mathbf{H}_{\text{out}\backslash i}(\mathbf{m}_{\text{in}} - \mathbf{m}_{\text{out}})\right]$
13:      **end for**
14: **end while**

---

An algorithm can be conveniently written by defining the following outer-loop quantities using the output $\boldsymbol{\lambda}_{\text{in}}$ of the inner loop,

$$\mathbf{g}_{\text{out}} \leftarrow \sum_{j=1}^{N} \mathbb{E}_{q_{\text{in}}}[\nabla\ell_j], \qquad \mathbf{H}_{\text{out}} \leftarrow \sum_{j=1}^{N} \mathbb{E}_{q_{\text{in}}}[\nabla^2\ell_j]. \tag{23}$$

We then set $\mathbf{m}_{\text{out}} \leftarrow \mathbf{m}_{\text{in}}$ and $\mathbf{S}_{\text{out}} \leftarrow \mathbf{S}_{\text{in}}$, and the corresponding natural parameter to be $\boldsymbol{\lambda}_{\text{out}}$. Using these, we can write the updates in the inner loop as follows (strictly in this order),

$$\begin{aligned}
\mathbf{g}_{\text{in}} &\leftarrow \mathbb{E}_{q_{\text{in}}}[\nabla\ell_i] - \mathbb{E}_{q_{\text{out}}}[\nabla\ell_i] + \mathbf{g}_{\text{out}}/N \\
\mathbf{H}_{\text{out}\backslash i} &\leftarrow \mathbf{H}_{\text{out}}/N - \mathbb{E}_{q_{\text{out}}}[\nabla^2\ell_i] \\
\mathbf{H}_{\text{in}} &\leftarrow \mathbb{E}_{q_{\text{in}}}[\nabla^2\ell_i] + \mathbf{H}_{\text{out}\backslash i} \\
\mathbf{S}_{\text{in}} &\leftarrow (1-\eta)\mathbf{S}_{\text{in}} + \eta N \mathbf{H}_{\text{in}} \\
\mathbf{m}_{\text{in}} &\leftarrow \mathbf{m}_{\text{in}} - \eta N \mathbf{S}_{\text{in}}^{-1}\left[\mathbf{g}_{\text{in}} + \mathbf{H}_{\text{out}\backslash i}(\mathbf{m}_{\text{in}} - \mathbf{m}_{\text{out}})\right]
\end{aligned} \tag{24}$$

These steps are implemented in Alg. 5 by using one Monte-Carlo sample to evaluate the expectations. We highlight in red the new parts added on top of SVRG. We note two useful points regarding the implementation: first, $\mathbf{S}_{\text{in}}$ need to be always updated before $\mathbf{m}_{\text{in}}$, and second, variance is further reduced if different example use different seeds (which is not explicitly written in the algorithm).

## B    DERIVATION OF THE ADAM-LIKE SVRG EXTENSION

To derive the SVRG extension of IVON, we will first write the VB objective in the form used by IVON; see Shen et al. (2024, Eq. 1). Essentially, they use mini-batches $\mathcal{B}$ of size B, and to accommodate this they scale the expected loss by a constant $\kappa\mathbb{E}_q[\ell_i]$. Setting $\kappa = N$ gives back the ERM loss but it can also be set to other values. In IVON, we also treat the regularizer $\ell_0$ explicitly by setting it to the weight decay. It is not merged in the losse $\ell_i$ as in the previous sections. With these changes, Sec. 3.2 can be written as where an explicit natural gradient of $\mathcal{L}_0$ is added,

$$\boldsymbol{\lambda}_{\text{in}} \leftarrow (1-\eta)\boldsymbol{\lambda}_{\text{in}} - \eta\kappa\left[\frac{1}{B}\sum_{i\in\mathcal{B}}\left(\widetilde{\nabla}\mathcal{L}_i(\boldsymbol{\lambda}_{\text{in}}) - \widetilde{\nabla}\mathcal{L}_i(\boldsymbol{\lambda}_{\text{out}})\right) + \frac{1}{\kappa}\sum_{j=1}^{N}\widetilde{\nabla}\mathcal{L}_j(\boldsymbol{\lambda}_{\text{out}}) + \frac{1}{\kappa}\widetilde{\nabla}\mathcal{L}_0(\boldsymbol{\lambda}_{\text{in}})\right].$$

We then plug in the natural parameter and natural gradients of $q = \mathcal{N}(\boldsymbol{\theta}|\mathbf{m}, \text{diag}(\mathbf{s})^{-1})$, which yields a Newton-like update very similar to Thm. 3.

To derive the IVON-PC update, we assume a quadratic regularizer $\ell_0 = s_0 \frac{1}{2} \boldsymbol{\theta}^\top \boldsymbol{\theta}$ with $s_0 > 0$. The VONcorr update can be written as follows where the new parts compared to Eq. 15 are in red,

$$\mathbf{s}_{\text{in}} \leftarrow (1 - \eta)\mathbf{s}_{\text{in}} + \eta\kappa \left[ \frac{1}{B} \sum_{i \in \mathcal{B}} \left( \mathbb{E}_{q_{\boldsymbol{\lambda}_{\text{in}}}}[\nabla^2 \ell_i] - \mathbb{E}_{q_{\boldsymbol{\lambda}_{\text{out}}}}[\nabla^2 \ell_i] \right) + \frac{1}{\kappa} \sum_{j=1}^{N} \mathbb{E}_{q_{\boldsymbol{\lambda}_{\text{out}}}}[\nabla^2 \ell_j] + \frac{s_0}{\kappa} \right],$$

$$\mathbf{m}_{\text{in}} \leftarrow \mathbf{m}_{\text{in}} - \eta\kappa \frac{1}{\mathbf{s}_{\text{in}}} \left[ \frac{1}{B} \sum_{i \in \mathcal{B}} \left( \mathbb{E}_{q_{\boldsymbol{\lambda}_{\text{in}}}}[\nabla \ell_i] - \mathbb{E}_{q_{\boldsymbol{\lambda}_{\text{out}}}}[\nabla \ell_i] \right) + \frac{1}{\kappa} \sum_{j=1}^{N} \mathbb{E}_{q_{\boldsymbol{\lambda}_{\text{out}}}}[\nabla \ell_j] + \frac{s_0}{\kappa} \mathbf{m}_{\text{in}} \right.$$

$$\left. + \left( \frac{1}{\kappa} \sum_{j=1}^{N} \mathbb{E}_{q_{\boldsymbol{\lambda}_{\text{out}}}}[\nabla^2 \ell_j] - \frac{1}{B} \sum_{i \in \mathcal{B}} \mathbb{E}_{q_{\boldsymbol{\lambda}_{\text{out}}}}[\nabla^2 \ell_i] \right) (\mathbf{m}_{\text{in}} - \mathbf{m}_{\text{out}}) \right]. \tag{25}$$

To write the update in IVON form, we make a few modifications.

1. For weight decay, we tune $\delta = s_0/\kappa$ directly.
2. We remove $\delta$ from the $\mathbf{s}_{\text{in}}$ update and divide the whole update by $\kappa$. The resulting update is written in terms of $\mathbf{h}_{\text{in}}$ such that $\mathbf{s}_{\text{in}} = \kappa(\mathbf{h}_{\text{in}} + \delta)$ and $\sigma_{\text{in}}^2 = 1/(\kappa(\mathbf{h}_{\text{in}} + \delta))$.
3. We use different learning rate for $\mathbf{m}_{\text{in}}$ and $\mathbf{h}_{\text{in}}$ updates. For $\mathbf{m}_{\text{in}}$, we use a scheduled $\eta_t$ for iteration $t$. For $\mathbf{h}_{\text{in}}$, we use $\beta_2 \in [0, 1)$.
4. We use momentum with learning rate $\beta_1 \in [0, 1)$ and debiasing for $\mathbf{g}$ (but not for $\mathbf{h}$).
5. We add a term for the update of $\mathbf{h}$ which ensures positivity of $\mathbf{h}$. We initialize $\mathbf{h}$ by a scalar constant $h_0 > 0$.
6. The scaling factor $\kappa$ is often set to $N$ but it can be different for cases when the effective number of examples is not immediately clear (for example, for LLM training).
7. We add an additional factor $\alpha$ in front of the outer gradients and Hessian, similarly to $\alpha$-SVRG (Yin et al., 2025).

## B.1 Handling Mega-Batches for Full-Gaussian

For full-Gaussians, we can build the site as follows,

$$\hat{q}_{\text{out}} \leftarrow \exp\left( -\boldsymbol{\theta}^\top \mathbf{g}_{\text{out}} - \tfrac{1}{2}(\boldsymbol{\theta} - \mathbf{m}_{\text{out}})^\top \mathbf{H}_{\text{out}}(\boldsymbol{\theta} - \mathbf{m}_{\text{out}}) \right),$$

$$\text{where} \quad \mathbf{g}_{\text{out}} \leftarrow \rho_1 \mathbf{g}_{\text{out}} + (1 - \rho_1) \sum_{i \in \mathcal{M}} \mathbb{E}_{q_{\text{out}}}[\nabla \ell_i] \tag{26}$$

$$\mathbf{H}_{\text{out}} \leftarrow \rho_2 \mathbf{H}_{\text{out}} + (1 - \rho_2) \sum_{i \in \mathcal{M}} \mathbb{E}_{q_{\text{out}}}[\nabla^2 \ell_i].$$

## C Ablations and Additional Results on Logistic Regression

Here, we run several ablations on the IVON-PC algorithm proposed in the main paper (Alg. 4).

**Extra term in $\mathbf{m}$-update.** The leftmost plot in Fig. 7 shows how the new term in the mean-update can cause instabilities in the training when naively implemented. This is likely due to the noisy reparametrization-trick based Hessian estimate. Using exact diagonal Hessian fixes the problem and leads to more stable training. Since the diagonal Hessian is expensive to compute, one can also simply downweigh the extra term by $0.01$. This works well throughout our experimnts. Therefore, we follow this approach in all logistic regression and image classification experiments. Aggressive gradient clipping ($\xi = 10^-4$) also somewhat stabilizes the training, but does not work as well as downweighting on this example. For more frequent refreshing as used in the transformer experiments, gradient clipping was enough to stabilize the training and no downweighting was required.

**Outer megabatch momentum.** We also perform two ablations over the outer momentum parameter $\rho$. The middle plot in Fig. 7 shows that when a few large outer batches are used, it is best to use $\rho = 0$ (no momentum) which always uses the freshest outer batch. In contrast, when using small megabatches as in the right plot in Fig. 7, larger values of $\rho = 0.9$ are advantageous.

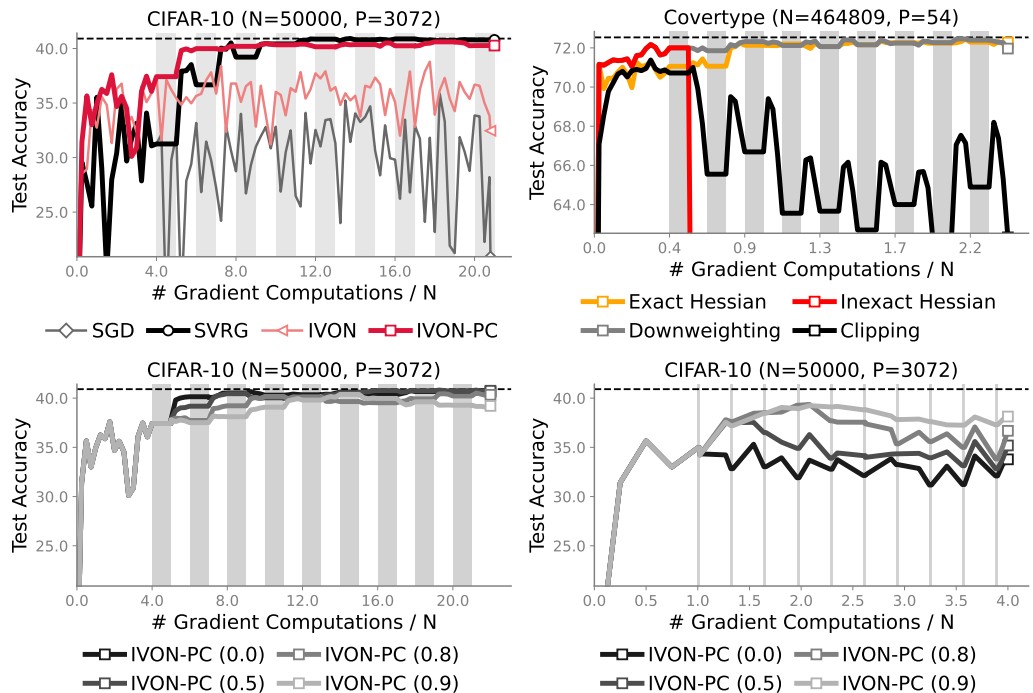

Figure 7: First plot shows additional result on CIFAR-10. For the second plot, we ablate the new red term in the **m**-update of Alg. 4 to show that it can be numerically unstable. Using exact Hessians or downweighting it improves the performance. Third plot shows that when large outer batches with infrequent refreshes are used, $\rho = 0$ performs best (no momentum). For small outer batches, small momentum works best ($\rho = 0.9$).

## D   HYPERPARAMETERS AND ADDITIONAL DETAILS

### D.1   IMAGE CLASSIFICATION WITH RESNETS

The ImageNet experiments run for 90 epochs, using batch size 384 on 8 GPUs which took around 12-15 hours. The learning rate is annealed to zero using a cosine schedule. For the CIFAR-10 experiments, we train for 100 epochs and batch size 50 on a single GPU, and each run took around 1-2 hours. As described in the main text, the learning rate is annealed to a quarter of the starting learning rate for all methods.

The hyperparameters for SGD, AdamW and IVON are set identical to the ones used in (Shen et al., 2024), except for CIFAR-10 where we use an ess of $5 \cdot 10^5$. All SVRG methods and IVON-PC inherit the hyperparameters from their base algorithm. The variance reduced methods use a megabatch size 50 times larger than the minibatch size on ImageNet and 10 times larger than the minibatch size on CIFAR-10. We tuned $\alpha$ for $\alpha$-SVRG and PC-IVON seperately. The optimal $\alpha$ is 0.2 for AdamW, 0.4 for SGD and 0.1 for IVON-PC. On ImageNet, we used $\alpha = 0.6$ but it was not overly tuned due to computational constraints. IVON-PC uses outer momentum of $\rho_1 = \rho_2 = 0.3$ on CIFAR-10 and no momentum was tried on ImageNet.

### D.2   PRETRAINING FROM SCRATCH

We first detail the experiments described in Sec. 4.3. We use the same set-up as in (Shen et al., 2024) which follows the nanoGPT repository found under https://github.com/karpathy/nanoGPT. We use an effective batch size of 480 achieved via 48 gradient accumulation steps to train a 125M parameter GPT-2 model from scratch on ca. 50B tokens from OpenWebtext in 100,000 steps. The AdamW run uses a learning rate of $6 \cdot 10^{-4}$, $(\beta_1, \beta_2) = (0.9, 0.95)$. The IVON run uses a learning rate of 0.3, $(\beta_1, \beta_2) = 0.9, 0.99999$, an effective sample size $\kappa = 1 \cdot 10^1 0$,

a Hessian initialization of $0.001$ as well as element-wise clipping of $0.001$. For IVON-PCM we use the same hyperparameters but starting at step 50,000 we add a correction with a megabatch that is 10 times the size of a minibatch and megabatch statistics that get updated after every 10 inner loop steps. We also use $\rho_1 = 0.6$ and $\rho_2 = 0.1$ for momentum. All these experiments are run on 8xA100 GPUs for up to one and a half days using bf16 and flash attention (Dao et al., 2022).

For the ablations we follow a similar recipe but rather use a smaller GPT-2 model which we downsize to ca. 33M parameters by using only 4 layers and 4 heads with an embedding dimension of 512. We run the experiments on wikitext103 and use the official train-validation splits for training and evaluation. Hyperparameters for IVON, IVON-PC, and AdamW are kept as above. We warmup IVON-PC and $\alpha$-SVRG with IVON and AdamW for 5,000 steps for the ablation over $\alpha$ and for 1,000 steps for the ablation over the refresh rate of the outer estimates. We use a batch size of 64 and a short context length of 128.

### D.3 CONTINUAL PRETRAINING

In this experiment we continually pretrain the GPT2-125M model from Shen et al. (2024) which is publicly available (`https://huggingface.co/team-approx-bayes/gpt2-small`) which was trained for 50B tokens on OpenWebText. All methods use a context length of 512 and a batch size of 80 and are trained on a single NVIDIA A100 80GB GPU with bf16 and flash attention to speed up training and reduce GPU memory utilization. We use the first 1B tokens from the Fineweb-edu-sample-10BT which is available openly under the following `https://huggingface.co/datasets/HuggingFaceFW/fineweb-edu` and split of the final 2000 documents for a validation split. For all experiments the learning rates are annealed to zero.

For IVON and IVON-PC, we continue using the optimizer state from pretraining but change some of the hyperparameters. We change the ess to $3 \cdot 10^{10}$ and use $\beta_2 = 0.9999$. The learning rate is set to $0.04$ instead of $0.3$, which was used for pretraining. With larger learning rates we found IVON to become less stable. For AdamW we start the optimizer state freshly and use a learning rate of $1 \cdot 10^{-4}$, down from the $6 \cdot 10^{-4}$ which was used for AdamW-trained models in Shen et al. (2024). Above, we found that loss sharply increased early in training which could lead to more forgetting of the data it was trained on. We use $\beta_1 = 0.9$ and $\beta_2 = 0.999$ which are oft-used for finetuning and the default choice in huggingface transformers (Wolf et al., 2020), which we use to implement our experiments.

For $\alpha$-SVRG and IVON-PC we warmstart training with 1,000 steps of AdamW and IVON, respectively, and use 40 inner steps before refreshing the outer gradient and Hessian estimates using 40 randomly sampled batches of size 80, i.e., 3200 examples and up to 1,638,400 tokens in total for estimating the gradients. We sample these batches randomly, so they need not overlap with the batches used in the following inner loop. We have found this to perform better in small experiments. Potentially, randomly sampling the batches reduces bias towards the same data used in the inner loop. For IVON-PC we set $\rho_1 = 0.3$ and $\rho_2 = 0.05$.

### D.4 FINETUNING

We finetune various models following the Transformer architecture (Vaswani et al., 2017). First, we use Vision Transformers (Dosovitskiy et al., 2021) with ca. 88M parameters. Our experiment uses OpenCLIP (Ilharco et al., 2021) and we only train the vision encoder but not the text encoder which produces label embeddings to which the image embeddings are matched. We train for 5 epochs on Cars (Krause et al., 2013), DTD (Cimpoi et al., 2014), GTSRB (Houben et al., 2013) and RESISC45 (Cheng et al., 2017) and start correction for IVON-PC after just 50 steps. We use a batch size of 8, 32 warmup steps, $\beta_1 = 0.9$, $\beta_2 = 0.99999$, the Hessian is initialized to $0.1$, ess equals $1 \cdot 10^{10}$, and the learning rate is initialized to $0.3$ and annealed to zero.

Next, we finetune two LLMs. First, we finetune the full Qwen2.5-0.5B-Instruct model on the first 50% of the training set of XSUM (Narayan et al., 2018) and evaluate on the corresponding test split. We finetune for a single epoch with a learning rate of 0.01, $\beta_1 = 0.9$, $\beta_2 = 0.99999$, ess of $1 \cdot 10^{10}$ and a Hessian initialized to $0.001$ as well as element-wise clipping to $0.001$. We use $\alpha = 0.7$ and refresh the outer gradients and Hessians every 50 steps. We use the same hyperparameters for LoRA-finetuning of LLAMA-3.1-8B but increase the learning rate to 0.05. We train the model for 3

epochs on GSM8k (Cobbe et al., 2021) and calculate the loss on both input and output tokens with a standard cross-entropy criterion. For all LLM methods we use greedy decoding and zero-shot prompting.

