# OpenReview forum: "SVRG and Beyond via Posterior Correction"
_ICLR.cc/2026/Conference — Submitted to ICLR 2026_

### Official Review · Reviewer_CxSC · 2025-10-21

**Soundness:** 4
**Presentation:** 3
**Contribution:** 1
**Rating:** 2
**Confidence:** 4

**Summary:**

The paper establishes a connection between SVRG (Johnson & Zhang, 2013) and posterior correction (PC)
(Khan, 2025), showing that SVRG can be understood as a special case of PC assuming isotropic Gaussian distributions.
PC is subsequently used to improve the IVON (Shen et al., 2024) optimizer across a wide range of experiments.

**Strengths:**

- The mathematical connection between SVRG and PC is novel.
- A strong and wide-ranging set of experiments together with a comprehensive set of ablations demonstrating the power of combining IVON with PC.

**Weaknesses:**

- While the equivalence of SVRG as a special case of PC is interesting, the paper reads like a
follow-up to Khan (2025) that demonstrates how PC and IVON can be combined to markedly improve
the performance of the latter, a comparison that was missing in Khan (2025).
The SVRG connection feels like an afterthought that does not provide deeper insights,
e.g., the extent to which convergence properties from SVRG still hold or can be extended to non-isotropic Gaussians, or the extent to which smoothness assumptions and variance reduction empirically hold.
- The paper lacks an analysis of the added computational cost, both from SVRG and from the Hessian approximations.
- All runs appear to be single runs, without any results on variance across training runs.
- No code is provided.

**Questions:**

- Q1: Does PC gain any theoretical properties from the derived SVRG connection?




_____
*Note: The low score is primarily due to the strong focus on SVRG without the paper gaining anything from this focus. If one were to ignore that and simply read it as an application paper combining IVON with PC and evaluating it extensively, the score would be higher.*

---

> ### Author Response · Authors · 2025-11-20
> **Thank you for your review!**
>
> We thank the reviewer for their review and for highlighting that “[t]he mathematical connection between SVRG and PC is novel” as well as the “strong and wide-ranging set of experiments”.
> We disagree though with their conclusion that the paper is a combination of IVON and PC with SVRG as an afterthought. We outline our stance below along with an address of their further points.
>
> > “The SVRG connection feels like an afterthought that does not provide deeper insights [...]”
> > “The low score is primarily due to the strong focus on SVRG without the paper gaining anything from this focus. If one were to ignore that and simply read it as an application paper combining IVON with PC and evaluating it extensively, the score would be higher.”
>
> We disagree with this opinion. The proposed connection does provide many deeper insights as described in the introduction (see line 39-49) and is not simply an afterthought. First, Khan (2025) mainly focused on adaptation of already-trained models, while we do this during training in a “double-loop” fashion, where an old posterior is built using an SVRG-like procedure and then $m$ stochastic inner steps are performed. Connected to Q1, this is fundamentally novel and is missing in Khan (2025). Second, the connection between SVRG and Bayes is nonexistent. Ours is the first work to make this connection. It should not be discarded as a simple “afterthought”. Third, there are several new deeper insights arising from this connection, as mentioned in the introduction.
>
> 1. (line 39) Novel extensions are automatically obtained by using more flexible exponential families. VSGD-PC and IVON-PC are just two examples.
>
> 2. (line 46) These new variants are not just IVON applied to perform PC, as the reviewer suggests, rather they are fundamentally new variants that are rare to see in the SVRG literature. For instance, the Hessian correction in IVON-PC is the first time such an algorithm is proposed using Bayes and applied at an LLM scale, as shown in the new result in Fig. 1, where we train GPT-2 (125M) from scratch on 50B tokens with improvements irrespective of when correction is added.
>
> 3. (line 41) Another deep insight is to connect variance-reduction methods to “knowledge transfer, for instance, through frequent mega-batch gradient computations” (L40-42).
>
> These insights are appreciated and outlined as a strength by all other reviewers, as well. While we do not provide any connections in terms of convergence guarantees, that is due to the fact that convergence guarantees of the naive Bayesian learning rule are also not yet established. This does not diminish the value of our work.
>
> We request the reviewer to reconsider their opinion and not discard our contributions as a mere application of existing ideas to SVRG.
>
> > “The paper lacks an analysis of the added computational cost, both from SVRG and from the Hessian approximations.”
>
> We have now added a dedicated section detailing the computational overhead (see Sec 3.4 in the revised version). Hessian estimation is obtained for free due to IVON’s use of the reparameterization trick (see line 4,8,9 in Algorithm 4).  Overheads for mega-batch refreshes (and corrections) are similar to standard SVRG and both require additional gradient and Hessian computation, which is the only cause of increase in computation overhead.
>
> > “No code is provided.”
>
> We will open-source all sourcecode upon accepted. For a preliminary version, please refer to our separate comment posted below.

---

> > ### Author Response · Authors · 2025-11-20
> > **Link to anonymized repository**
> >
> > Please find below an anonymized repository that can be used to reproduce results with our method on image classification and language modeling tasks:
> >
> > https://anonymous.4open.science/r/svrgandbeyondiclr26/

---

> > > ### Author Response · Authors · 2025-11-27
> > > **Gentle Reminder.**
> > >
> > > Dear Reviewer,
> > >
> > > Thank you again for your effort!
> > >
> > >
> > > Since the discussion period will be closing soon, please let us know if there are any further questions.

---

### Official Review · Reviewer_cZ3B · 2025-10-31

**Soundness:** 3
**Presentation:** 3
**Contribution:** 3
**Rating:** 4
**Confidence:** 3

**Summary:**

The paper introduces a connection between Stochastic Variance Reduced Gradient (SVRG) with Posterior Correction (PC). The authors show that SVRG emerges as a special case of posterior correction when using isotropic Gaussian distributions. This insight allows them to derive new SVRG variants by applying posterior correction to broader exponential-family distributions. Two extensions are proposed: 1. Newton-like variant (VON-PC) — incorporates stochastic variance reduction for both gradients and Hessians, improving stability and enabling second-order corrections. 2. Adam-like variant (IVON-PC/IVON-PCM) — adapts posterior correction over the IVON optimizer, showing strong performance in continual pre-training and fine-tuning of large models like GPT-2 and ViT.

**Strengths:**

1. The original contribution by establishing a connection between SVRG and posterior correction. The interpretation reframes variance reduction as a form of knowledge transfer, a new perspective that unifies two previously separate research threads.
2. The authors successfully extend this connection to derive two new SVRG variants: One is a Newton-like method incorporating stochastic variance reduction for both gradients and Hessians, introducing Hessian corrections rarely explored in prior work; another is an Adam-like method (IVON-PC/IVON-PCM) with diagonal covariance approximations.

**Weaknesses:**

1. Compared with the well established optimization method e.g., AdamW, posterior correction does not yield clear improvements in deep learning tasks.
2. Can the variance reduction method be applied to reinforcement learning? The authors can explore the experiments on either RLVR of LLMs post-training or some other traditional tasks in RL area. It can also compared with TRPO (Trust Region Policy Optimization).

**Questions:**

How about its computational overhead (e.g., Hessian estimation, mega-batch refresh) ?

---

> ### Author Response · Authors · 2025-11-20
> **Thank you for your review!**
>
> We thank the reviewer for their review as well as for emphasizing that it establishes "a new perspective that unifies two previously separate research threads”. We have added a new result in Fig. 1 which addresses the point about “no clear improvements” compared to Adam. We have added Sec 3.4 which discusses the computational overhead. We hope that the reviewer will reconsider their score. Detailed comments are below.
>
> > “Compared with the well established optimization method e.g., AdamW, posterior correction does not yield clear improvements in deep learning tasks.”
>
> Thanks for your feedback. We believe this remark arises due to Fig. 1 which is a bit misleading. We have now revised Fig. 1 where clear boosts are seen both over Adam and IVON when training GPT2 (125M) on 50B tokens. IVON-PCM achieves a lower perplexity of 17.4 compared to 18.4 by AdamW and 18.0 by IVON. The improvements are consistent, irrespective of the iteration when the corrections are started (see the right panel). This is unlike any existing result in the SVRG literature where variance reduction is expected to have the most impact in the beginning of the training. It also far exceeds the scale of any prior work on SVRG that we are aware of.
>
> We would like to point out that we also get faster convergence than SVRG in convex settings (Fig. 3), so we do improve cases where SVRG already works. We would like to point that almost *all* SVRG methods have never delivered stronger empirical results for deep learning. This is mentioned in line 34-35. Our results focus on its potential for “non-traditional” settings where we do see promising (even though not conclusive) results.
>
> > “Can the variance reduction method be applied to reinforcement learning? The authors can explore the experiments on either RLVR of LLMs post-training or some other traditional tasks in RL area. It can also compared with TRPO (Trust Region Policy Optimization).”
>
> Yes, our method is an optimization algorithm and can easily be used with any common reinforcement learning algorithm, including but not limited to TRPO. The only requirement is that there is an (estimate of) first order gradients. We do not believe this to be a weakness but rather a strength of our method and are happy to explore further settings, such as, RL in future work.
>
> > How about its computational overhead (e.g., Hessian estimation, mega-batch refresh) ?
>
> Hessian estimation is obtained for free, thanks to IVON’s use of the reparameterization trick (see lines 4,8,9 in Algorithm 4).  Overheads for mega-batch refreshes (and corrections) are similar to standard SVRG and both require additional gradient and Hessian computation (which is the only cause of increase in computation overhead). We have now added a dedicated section detailing the computational overhead (see Sec 3.4 in the revised version).

---

> > ### Author Response · Authors · 2025-11-27
> > **Gentle Reminder.**
> >
> > Dear Reviewer,
> >
> > Thank you again for your effort!
> >
> > Since the discussion period will be closing soon, please let us know if there are any further questions.

---

### Official Review · Reviewer_XyZw · 2025-10-31

**Soundness:** 3
**Presentation:** 3
**Contribution:** 3
**Rating:** 6
**Confidence:** 3

**Summary:**

The paper establishes a connection between SVRG and posterior correction, enabling the generalization of variance reduction method to the IVON optimizer. At a high level, it aims to establish a relationship between non-Bayesian optimization methods and variational inference within a common theoretical framework.

**Strengths:**

- The connection between SVRG and posterior correction is sound.
- An advantage of the connection is that it generalizes the variance reduction method to higher-order optimizers, leading to a novel IVON-PC.

**Weaknesses:**

- While the theoretical analysis provides a fresh perspective on SVRG, the resulting IVON-PC method does not appear to offer practical benefits.

  - Figure 3 shows an initial improvement, but IVON-PC ultimately requires a similar number of gradient computations as SVRG to reach comparable final performance. Likewise, Figure 5 demonstrates that their final performance remains nearly identical.

  - The gains reported in Table 5 are also insignificant, especially when considering the error bars.

- Another limitation is that the established connection does not clarify why SVRG or the proposed IVON-PC fail to deliver stronger empirical results.

The above two points raise questions about whether we obtained any value through the established connection. Nonetheless, I lean toward acceptance, as the work introduces a novel viewpoint and may inspire deeper future investigations on SVRG.

**Questions:**

-

---

> ### Author Response · Authors · 2025-11-20
> **Thank you for your review!**
>
> We thank the reviewer for their review and for pointing out that “[t]he connection between SVRG and posterior correction is sound” and “generalizes the variance reduction method to higher-order optimizers, leading to a novel IVON-PC”. We have updated Fig. 1 to show a result training GPT2 (125M) on 50B tokens and clarified a possible confusion in Table 1 where indicated gains (with a plus sign) might have been mistaken for error bars. We hope that this addresses all concerns and that the reviewer can reconsider their valuation of our paper.
>
> > “While the theoretical analysis provides a fresh perspective on SVRG, the resulting IVON-PC method does not appear to offer practical benefits. Another limitation is that the established connection does not clarify why SVRG or the proposed IVON-PC fail to deliver stronger empirical results.”
>
> While we understand your criticism, we would like to point out that this is true for *all* SVRG methods, not just our new proposal (see line 34-35). None of the methods have ever offered practical benefits or delivered stronger empirical results for deep learning. Here, we explored the potential for “non-traditional” settings which are not common in the SVRG literature. To make the point clearer, we have now revised Fig. 1 where clear boosts are seen for GPT2 (125M) training on 50B tokens. The improvements are consistent, irrespective of the iteration when the corrections are started (see the right panel). This is unlike any existing result in the SVRG literature where variance reduction is expected to have the most impact in the beginning of training. It also far exceeds the scale of any prior work on SVRG that we are aware of.
>
> While it does not *directly* explain why SVRG fails, it does so indirectly, by going beyond SVRG and applying the idea in non-traditional settings, as promised. Note that we do get faster convergence than SVRG in convex settings (Fig. 3), so we do improve cases where SVRG already works. But we also hope that our fresh new perspective will open new ways to improve SVRG, and with this perspective we believe our results to offer practical benefit and deliver useful results. However, we are happy to continue this discussion with you if there are ways that we can further improve our presentation.
>
> > “The gains reported in Table 5 are also insignificant, especially when considering the error bars.”
>
> Perhaps there is a typo here because we do not have a Table 5. Do you mean Table 1, or perhaps Figure 5? Both do not have error bars. It is possible that, in Table 1, the improvements in accuracy (indicated in green) are mistaken to be error bars. In the revised version, we have made this point clear in the caption.

---

> > ### Comment · Reviewer_XyZw · 2025-11-24
> > **Response to Authors**
> >
> > I appreciate the authors’ response.
> >
> > - I was referring to Table 1. The gains presented there remain minor, and it’s difficult to draw a meaningful conclusion about one method being better than another.
> >
> > - The newly added Figure 1 shows the same pattern as my earlier comment on Figure 3. The initial improvement is promising, but IVON-PCM ultimately requires roughly the same number of gradient evaluations as IVON or AdamW to reach similar final performance.
> >
> > Despite the limitations, I agree with the authors that this fresh perspective is interesting, which may motivate further investigations on SVRG. I stand with my initial rating.

---

> ### Author Response · Authors · 2025-11-24
> **Thank you for your response!**
>
> Thank you very much for your response!
>
> We would like to point out though that Fig. 3 does show a better final performance for IVON-PCM than IVON, perhaps you are referring to Fig. 4 and 5?
>
> > The newly added Figure 1 shows the same pattern as my earlier comment on Figure 3. The initial improvement is promising, but IVON-PCM ultimately requires roughly the same number of gradient evaluations as IVON or AdamW to reach similar final performance.
>
> In Fig. 3, IVON is significantly worse in the case of logistic regression and IVON-PCM converges both to a better final value and even faster than SVRG which we believe is a strong result. Perhaps you are referring to Fig. 4 and 5? There, we show traditional deep learning settings on CIFAR-10 and ImageNet where SVRG-style methods do not give a direct benefit when matching the number of examples seen.

---

### Meta-Review · Area_Chair_WSLt · 2026-01-07

**Summary:**

The paper establishes a connection between Stochastic Variance Reduced Gradient (SVRG) and Posterior Correction (PC), showing that SVRG is a special case of PC when using an isotropic Gaussian distribution. Consequently, authors use this connection to propose a broader variant of SVRG based on the exponential family. In concrete terms, existing optimisers (e.g., IVON) are extended with PC to yield improved stability and performance, as demonstrated by comprehensive experiments.

Reviewers have expressed borderline/negative opinions about this submission, with the main criticism that the proposal is too incremental, has weak empirical improvements, and that evaluation is based only on a single run. Further concerns have been about the additional cost of the method and a lack of code availability, which both has been addressed by the authors during the rebuttal.

**Reviewer Concerns:**

Some minor concerns have been addressed during the rebuttal. The main remaining concerns are that the approach is too incremental and shows only weak empirical improvements, as indicated by inspection of the discussion and the updated paper, which includes experimental results (e.g., Fig. 1) that aim to highlight the method's success but only show limited performance gains. Moreover, as the results are only shown for a single run (as pointed out by CxSC), a clear judgment of how much the proposal actually improves results is challenging.

**Reviewer Scores:**

The revised scores are expected to be as follows:

- [XyZw]: Mentioned to have kept the score. => 6
- [cZ3B]: Likely to have kept or increased the score. => 4/5
- [CxSC]: Likely to have kept or increased the score. => 2/3

---

### Decision · Program_Chairs · 2026-01-26

Reject